# A Cellulose-Derived Nanofibrous MnO_2_-TiO_2_-Carbon Composite as Anodic Material for Lithium-Ion Batteries

**DOI:** 10.3390/ma14123411

**Published:** 2021-06-20

**Authors:** Shun Li, Ming Yang, Guijin He, Dongmei Qi, Jianguo Huang

**Affiliations:** 1Department of Chemistry, Zhejiang University, Hangzhou 310027, China; 21937064@zju.edu.cn (M.Y.); guikinghe@zju.edu.cn (G.H.); 2School of Engineering, Zhejiang Agriculture and Forestry University, Hangzhou 311300, China; 3Analysis Center of Agrobiology and Environmental Sciences, Zhejiang University, Hangzhou 310027, China; qidongmei@zju.edu.cn

**Keywords:** bio-inspired materials, cellulosic substances, titania film, MnO_2_ nanosheets, lithium-ion batteries

## Abstract

A bio-inspired nanofibrous MnO_2_-TiO_2_-carbon composite was prepared by utilizing natural cellulosic substances (e.g., ordinary quantitative ashless filter paper) as both the carbon source and structural matrix. Mesoporous MnO_2_ nanosheets were densely immobilized on an ultrathin titania film precoated with cellulose-derived carbon nanofibers, which gave a hierarchical MnO_2_-TiO_2_-carbon nanoarchitecture and exhibited excellent electrochemical performances when used as an anodic material for lithium-ion batteries. The MnO_2_-TiO_2_-carbon composite with a MnO_2_ content of 47.28 wt % exhibited a specific discharge capacity of 677 mAh g^−1^ after 130 repeated charge/discharge cycles at a current rate of 100 mA g^−1^. The contribution percentage of MnO_2_ in the composite material is equivalent to 95.1% of the theoretical capacity of MnO_2_ (1230 mAh g^−1^). The ultrathin TiO_2_ precoating layer with a thickness ca. 2 nm acts as a crucial interlayer that facilitates the growth of well-organized MnO_2_ nanosheets onto the surface of the titania-carbon nanofibers. Due to the interweaved network structures of the carbon nanofibers and the increased content of the immobilized MnO_2_, the exfoliation and aggregation, as well as the large volume change of the MnO_2_ nanosheets, are significantly inhibited; thus, the MnO_2_-TiO_2_-carbon electrodes displayed outstanding cycling performance and a reversible rate capability during the Li^+^ insertion/extraction processes.

## 1. Introduction

Lithium-ion batteries (LIBs) are regarded as one of the most practical and effective technologies for the development of electric vehicles, mobile devices, and reproducible energy integration [1,2]. The key points to boost the power and energy density of LIBs largely rely on the physiochemical properties of the anodic and cathodic materials [3,4]. Traditionally adopted commercial graphite anode materials deliver a relatively low specific capacity of 372 mAh g^−1^, which are unable to satisfy the increasing requirements for high energy storage [5]. Compared with conventional commercialized graphite anodes, transition metal oxides have caused a great deal of interest thanks to their high theoretical capacities and reliable discharging rates [6,7,8,9,10,11]. Among the various transition metal oxides, manganese oxide (MnO_2_) is an attractive electrode candidate for LIBs due to its high storage capacity (1230 mAh g^−1^), rich abundance, and environmental friendliness [12,13]. However, the practical applications of MnO_2_-based anode materials are greatly limited by their poor intrinsic electric conductivity (~10^−7^–10^−8^ S cm^−1^) and severe volume expansion and pulverization of MnO_2_ matter from repeated charge/discharge cycles [14,15].

To overcome these issues, many attempts have been made to increase the capacity stability and the electric conductivity of MnO_2_-based anode materials. An effective approach is the construction of nanoarchitectures with new composites composed of electrically conductive carbon or graphite and MnO_2_ so as to enhance the electrochemical performance of the anode materials [16]. For example, a curly MnO_2_ nanoflake electrode delivered a reversible capacity of 696 mAh g^−1^ as the cycling rate returned to 50 mA g^−1^, which was ascribed to its high surface-to-volume ratio and stable structure [17]. A coaxial MnO_2_/CNT array composite material delivered a first discharge capacity of ca. 2000 mAh g^−1^ at a current rate of 50 mA g^−1^. Unfortunately, the capacity suddenly dropped to 500 mAh g^−1^ after 15 cycles. The fast degradation of the MnO_2_/CNT materials was due to the structural destruction of the thick MnO_2_ layer deposited on the outer surface of the CNTs [18]. It has also been demonstrated that carbon nanohorns (CNHs) possess good electric conductivity and a large surface area and acted as a carrier for MnO_2_ nanosheets; the composite showed a reversible discharge capacity of 565 mAh g^−1^ as tested at a 0.1-C current rate (1 C = 1000 mA g^−1^) after 60 charge/discharge cycles [19]. Moreover, a layer-by-layer graphene-MnO_2_ nanotube composite exhibited a reversible capacity of 495 mAh g^−1^ after 40 cycles at a rate of 100 mA g^−1^ [20]. However, these MnO_2_-based electrode materials showed rapid capacity degradation and failed to realize satisfactory cycling performance and reversible rate capability.

TiO_2_ has been explored as an active anodic material for LIBs owing to its attractive properties, such as its low cost, safety, and relatively low volume expansion during the charge/discharge processes. The excellent physicochemical characteristics of TiO_2_ lead to the long cycling life and durability of the electrodes. However, the low theoretical capacity of around 335 mAh g^−1^ and the inherently slow transport kinetics for both Li ions and electrons severely hinder TiO_2_’s practical application [21,22]. Various nanostructured TiO_2_ and TiO_2_-based carbon composites have been developed, aiming at improving the ionic and electric transportation of electrodes during Li/Li^+^ processes [23]. A bio-inspired multilevel nanofibrous silver-nanoparticle-anatase-rutile-titania composite material, which was composed of anatase-phase titania nanotubes with rutile-phase titania and further immobilized with silver nanoparticles, displayed a stable capacity of 120 mAh g^−1^ after 100 cycles, with a Coulombic efficiency of approximately 100% [24]. Moreover, it was also confirmed that nanofibrous TiO_2_-coated carbon composites deposited with silver nanoparticles exhibited an initial discharge capacity of 1323 mAh g^−1^ and maintained 320 mAh g^−1^ in the first 150 cycles when employed as anodes for LIBs [25]; however, these TiO_2_-related anode materials failed to achieve a high capacity and capacity retention.

The biomimetic synthesis approach is believed to be an effective way to fabricate functional nanomaterials with specific structure morphologies and to design physicochemical properties that are faithfully inherited from an original biological organism [26,27,28,29,30]. Natural cellulose substances, which possess unique interwoven network structures, are verified to be a desirable structural matrix and template for the self-assembly of active guest components on their surfaces through a layer-by-layer method, which shows great potential in the fabrication of cellulose-derived materials for lithium-ion batteries. The hierarchical porous structures of cellulose substance-derived functional nanomaterials are unique [31], and cannot be obtained using other synthesis method, such as the combination of different nanoscale building blocks [16]. Herein, a bio-inspired nanofibrous MnO_2_-TiO_2_-carbon composite was prepared by employing natural cellulose substances (commercial laboratory filter paper) as both the carbon source and the structural matrix using a facile sol–gel process and hydrothermal procedures. The carbon nanofibers were deposited on an ultrathin TiO_2_ layer and additional MnO_2_ nanosheets were then immobilized to construct the MnO_2_-TiO_2_-carbon nanoarchitecture. The three-dimensional network structure of the composite material was fully inherited from the initial cellulosic substances. In addition, the titania precoating interlayer greatly contributed to the well-organized immobilization, as well as to the higher loading of content of the porous MnO_2_ nanosheets. The MnO_2_-TiO_2_-carbon material displayed preeminent cycling stabilities and reversible rate capabilities when utilized as an anodic material for LIBs. The improvement of the electrochemical capabilities of the nanocomposite were ascribed to the distinct interconnected structures and the high surface-to-volume ratio of the carbon matrix, which significantly alleviated the volumetric expansion and agglomeration of the anode material upon cycling. Moreover, the carbon scaffold also acted as a conductive matrix and facilitated fast Li^+^ and ion transport in the Li/Li^+^ processes. The titania thin film also contributed to the strong adhesion between the carbon nanofiber and the MnO_2_ nanosheets, and suppressed the pulverization and exfoliation of the MnO_2_ nanosheets from the MnO_2_-TiO_2_-carbon material. Due to the synergetic effect of the three components, the MnO_2_-TiO_2_-carbon anode material exhibited improved electrochemical performance.

## 2. Materials and Methods

### 2.1. Materials

Ordinary quantitative ashless filter paper was obtained from Hangzhou Xinhua Paper Industry Co. Ltd. (Hangzhou, China). Potassium permanganate (KMnO_4_, >99%) was bought from Sigma Aldrich (St. Louis, MO, USA). Other chemical reagents used in the experiments were acquired and employed without additional treatment. DI water was utilized in all fabrication processes and was purified using a Milli-Q Advantage A10 system (Millipore, Bedford, MA, USA) with a given electrical resistance of over 18.2 MΩ cm.

### 2.2. Fabrication of the Nanofibrous MnO_2_-TiO_2_-Carbon and MnO_2_-Carbon Composites

In a representative procedure, 5-layer titania gel film was deposited onto the surface of cellulose nanofibers from bulk quantitative filter paper using a surface sol–gel method (Scheme 1a,b), which was followed by carbonization of the as-synthesized TiO_2_ gel film by precoating the cellulose nanofiber in an Ar atmosphere at 450 °C for 6 h to obtain the TiO_2_-carbon nanocomposite (Scheme 1c) [25]. Subsequently, 20.0 mg of TiO_2_-carbon material was added into 20 mL DI water with sonification for 1.5 h, in order to control the MnO_2_ content in the resultant composite. Various weights of KMnO_4_ (30 and 25 mg) were then added and stirred under the ambient temperature for 5 h. Finally, the two as-obtained violet mixtures were transferred to 50-mL Teflon-lined autoclaves (Zhongnuo Instrument Co., Ltd., Xi’an, China), respectively, and further heated in a muffle furnace (Jinghong Experimental Equipment Co., Ltd., Shangai, China) at 150 °C for 5 h. After cooling to an ambient temperature, the brownish products were rinsed with DI water and absolute ethanol 3–4 times, and were dried further overnight in an electric vacuum oven (80 °C, Yiheng Scientific Instrument Co., Ltd., Shangai, China) (Scheme 1d). As verified by thermal gravimetric analyses (TGA), the contents of MnO_2_ in the MnO_2_-TiO_2_-carbon composites were measured to be 47.28 wt % and 37.81 wt %, respectively. The corresponding composites were designated as MnO_2_-TiO_2_-carbon-47.28% and MnO_2_-TiO_2_-carbon-37.81%, respectively. As for the control experiment, MnO_2_-carbon materials with varying MnO_2_ contents were fabricated under same conditions without a pre-deposited titania gel layer (Scheme 1e,f). The TGA measurements showed that the MnO_2_ contents in the MnO_2_-carbon composite were 40.15 wt % and 33.30 wt % when the addition of KMnO_4_ was 30 and 25 mg, respectively. The obtained composites were designated as MnO_2_-carbon-40.15% and MnO_2_-carbon-33.30%, respectively. The MnO_2_ nanoparticles (MnO_2_-NPs) were fabricated by the addition of 0.2 g KMnO_4_ and 0.2 mL H_2_SO_4_ (95 wt %) to 25 mL DI water to form a precursor, and followed by hydrothermal reaction at 150 °C for 4 h.

### 2.3. Characterizations

A small amount of specimen was dispersed in absolute ethanol followed by ultrasonication for 60 s; then, the suspensions were dropped onto aluminum foil for field emission scanning electron microscopy (FE-SEM) tests on a Hitachi SU-8010 (HITACHI, Tokyo, Japan) instrument with an acceleration voltage of 5.0 kV. In the meantime, the resultant mixtures were placed on a copper grid for transmission electron microscopy (TEM) observations on a Hitachi HT-7700 (HITACHI, Tokyo, Japan) tool with an acceleration voltage of 100 kV. High-resolution transmission electron microscopy (HR-TEM) and selected area electron diffraction (SAED) measurements were conducted on a JEM-2100F apparatus (JEOL, Tokyo, Japan) with an acceleration voltage of 300 kV. X-ray diffraction (XRD) measurements were performed on a Philips X’Pert PRO diffractometer (PANalytical B.V., Alemlo, The Netherlands) using Cu_Kα_ (λ = 0.15405 nm) as a radiant source. X-ray photoelectron spectra (XPS) were detected by employing a VG Escalab Mark 2 spectrophotometer (VG Instruments, Manchester, UK) with a MgK_α_ X-ray source (1253.6 eV). High-resolution scans of Mn were performed at 0.2 eV increments (sweep time: 1000 ms eV^−1^, each region: 30 energy sweeps). The C 1s peak at 284.50 eV was used as a standard for all the XPS peaks. Raman spectra were obtained using a Jobin Yvon LabRam HR UV Raman spectrometer (HORIBA, Paris, France) with an excitation wavelength of 532 nm. Thermal gravimetric analyses (TGA) were performed on a Mettler Toledo STARe System TGA2 (Mettler Toledo Crop., Zurich, Switzerland) at a heating rate of 10 °C min^−1^, from room temperature to 800 °C in a nitrogen atmosphere. The specific surface areas of the samples were determined through measurement of the N_2_ adsorption–desorption isotherms at −196 °C with a Micromeritics ASAP-2020 tool (Micromeritics, Norcross, GA, USA), and the Brunauer–Emmett–Teller (BET) method was applied to compute the surface area and pore volume of the MnO_2_-TiO_2_-carbon composites.

### 2.4. Electrochemical Measurements

The working electrodes were prepared by mixing Super P (10 wt %) and polyvinylidene fluoride binder (PVDF, 10 wt %) with the active material (80 wt %) in *N*-methyl-2-pyrrolidinone (NMP) solvent. The slurries were uniformly coated onto copper foam and dried in a vacuum oven (80 °C, 12 h); the electrodes were then pressed at 12 atm and the mass loading of the active material was ca. 1.5–2.0 mg for each electrode. Coin-type cells (CR2025, DodoChem, Suzhou, China) were assembled in an argon filled glove box with concentrations of oxygen and water molecules less than 0.1 ppm. A Li metal plate was used as the auxiliary electrode and polypropylene (Celglad 2300, DodoChem, Suzhou, China) film was used as the separator. The electrolyte was composed of 1.0 M LiPF_6_ in ethylene carbonate (EC), ethyl methyl carbonate (EMC), and dimethyl carbonate (DMC) (EC:EMC:DMC, v:v:v = 1:1:1). Cyclic voltammetry (CV) tests were performed on a CHI760D electrochemical workstation (CH instruments, Shanghai, China) with a scan rate of 0.1 mV s^−1^ and voltage window of 0.01–3.0 V. Galvanostatic charge/discharge performances were tested using a Neware battery testing system (Neware Technology Co., Ltd., Shenzhen, China) in the voltage range from 0.01 to 3.0 V (Li/Li^+^) at ambient temperature. Electrochemical impedance spectra (EIS) measurements were recorded in the frequency range of 100 kHz to 0.01 Hz. The structural stability of the electrode composites after 200 charge/discharge cycles was confirmed via electron microscope observations.

## 3. Results and Discussion

### 3.1. Structural Characterizations of the MnO_2_-TiO_2_-Carbon and MnO_2_-Carbon Composites

As illustrated in Scheme 1, the MnO_2_-based carbon composites were obtained by utilizing natural cellulosic substances (e.g., common commercial filter paper) as both the scaffolds and carbon sources. The MnO_2_ nanosheets were uniformly immobilized on the surfaces of the ultrathin TiO_2_-coated carbon nanofibers or on the nanofibrous carbon to acquire MnO_2_-TiO_2_-carbon (Scheme 1a–d) and MnO_2_-carbon composites (Scheme 1a–f). The resultant materials with varied MnO_2_ contents were denoted as MnO_2_-TiO_2_-carbon-47.28%, MnO_2_-TiO_2_-carbon-37.81%, MnO_2_-carbon-40.15%, and MnO_2_-carbon-33.30%, respectively.

The morphology and structure properties of the MnO_2_-TiO_2_-carbon-47.28% composite are shown in Figure 1. It can be clearly seen that the composite is composed of interconnected microfibers, and each microfiber consist of nanofiber assemblies with a diameter range from tens of to several hundred nanometers (Figure 1a). The three-dimensional network structures of the nanocomposite were perfectly inherited from the original cellulose substances, and the closely-packed MnO_2_ sheets (thickness ca. 20 nm) were evenly immobilized on the surfaces of the titania-coated carbon nanofibers (Figure 1b). The diameter of an individual MnO_2_-TiO_2_-carbon-47.28% nanofiber isolated from the nanofiber assemblies was ca. 400 nm (Figure 1c). A high-magnification TEM image (Figure 1d) of the composite nanofiber shows MnO_2_ nanosheets with a pore structure that was formed from the evaporation of the H_2_O molecules.

The selected area electron diffraction (SAED) of an individual MnO_2_-TiO_2_-carbon-47.28% nanofiber and the relevant EDS elemental mapping of carbon (C), titanium (Ti), oxygen (O), manganese (Mn) are shown in Figure 2. It was observed that C appears in the central area of the composite nanofiber, whereas the distribution zones of Mn and O were comparable and wider than those of Ti and C. This implies that the MnO_2_ nanosheets were uniformly immobilized on the surface of the titania-coated carbon nanofibers. In comparison, the multilevel structures of original filter paper were perfectly sustained by the resultant MnO_2_-TiO_2_-carbon-37.81% composite (Appendix A, Appendix A). The densely loaded MnO_2_ nanosheets (thickness ca. 15 nm) were anchored to the surface of the titania-carbon nanofibers, and MnO_2_ nanorods with a diameter of around 20 nm were connected by the adjacent MnO_2_-TiO_2_-carbon-37.81% composite nanofibers to construct three-dimensional cross-linked network structures. When these two composites were tested as anode materials for LIBs, the MnO_2_-TiO_2_-carbon-47.28% composite showed a higher electrochemical performance than the sample MnO_2_-TiO_2_-carbon-37.81%. This was attributed to the stronger binding force between the well-organized MnO_2_ nanosheets and the titania coating layer in MnO_2_-TiO_2_-carbon-47.28%. Hence, the large volume change and pulverization of the MnO_2_ nanosheets were significantly alleviated.

Figure 3 shows SEM and TEM images of the MnO_2_-carbon-40.15% composite, extensive MnO_2_ nanosheets and a few MnO_2_ nanorods grew on the surfaces of the nanofibrous carbon (Figure 3a,b). The immobilized MnO_2_ nanosheets with a porous structure can obviously be seen in Figure 3c,d. In contrast, the density of the MnO_2_ nanosheets in MnO_2_-carbon-40.15% was lower than that of the MnO_2_-TiO_2_-carbon-47.28% composite because the titania film provided more active sites for the growth of MnOOH, which resulted in the well-organized growth of MnO_2_ nanosheets of smaller sizes [32]. The original multilevel morphology of the structures can still be observed in the MnO_2_-carbon-33.30% materials (Appendix A), showing that the closely packed MnO_2_ nanosheets are immobilized on the surface of carbon nanofibers, crosslinked MnO_2_ nanorods are intertwined among the MnO_2_-carbon-33.30% nanofibers surfaces. However, the as-fabricated MnO_2_-NPs with a mean diameter of ca. 30 nm and length of hundreds of nanometers tended to aggregate (Appendix A).

Energy-dispersive X-ray spectrometry (EDS) mapping test results are exhibited in Figure 4 to further verify the structural and elemental construction of the MnO_2_-carbon-40.15% materials. The uniformly distributions of Mn and O elements indicate the successful immobilization of MnO_2_ onto the surface of the carbon nanofiber (Figure 4d–f). The growth of MnO_2_ in the MnO_2_-TiO_2_-carbon and MnO_2_-carbon composites are based on the following reaction [33]:4MnO_4_^−^ + 3C + H_2_O = 4MnO_2_ + CO_3_^2−^ + 2HCO_3_^−^(1)

In this process, the carbon can easily react with KMnO_4_ in solutions and form a strong adhesion force between the MnO_2_ sheets and the titania-coated carbon or the carbon nanofibers. The content of MnO_2_ in the MnO_2_-TiO_2_-carbon-47.28% exceeded that of the MnO_2_-carbon-40.15% composite, which was due to the protective effects of the titania precoating layer. The interconnected carbon nanofibers derived from the natural cellulose substance served as a conductive matrix to enhance electron transportation and shorten the pathway of Li^+^ during cycling. In addition, the ultrathin TiO_2_ coating layer well facilitated the growth of MnO_2_ nanosheets on the surface of TiO_2_-carbon nanofibers through a redox reaction, as mentioned above.

To determine the phase structure of the composite materials, Figure 5a shows the X-ray diffraction results of the corresponding samples; the major peaks presented in the MnO_2_-TiO_2_-carbon-47.28%, 37.81%, and MnO_2_-carbon-40.15%, 33.30% materials are indexed to the (001), (002), (−111) and (020) planes of *α*-MnO_2_ crystal (JCPDS#. 80-1098) [34]. A broad diffraction peak situated at approximately 26° corresponded to the (002) plane of graphitic carbon in the TiO_2_-carbon composite, and the relatively weak peak intensity of the carbon reflection was seen in MnO_2_-based carbon composites [35]. The XRD pattern of the MnO_2_-NPs was attributed to the monoclinic potassium birnessite, which was composed of 2D edge-shared MnO_6_ octahedral layers with K^+^ and H_2_O molecules contained in the interlayer space [4].

The structure features of the MnO_2_-TiO_2_-carbon composites were further verified using Raman spectra, as shown in Figure 5b. A broad peak located around 650 cm^−1^ was assigned to the Mn-O stretching vibration of the MnO_6_ octahedra [33]. Two weak peaks positioned at 1350 and 1590 cm^−1^ were ascribed to the D- and G-bands of the carbon materials, respectively [36]. The corresponding peak intensities of the D- and G-bands of carbon components became lower with the increasing content of MnO_2_ nanosheets in the MnO_2_-TiO_2_-carbon and MnO_2_-carbon composites, which was due to the depletion of carbon during the formation of the MnO_2_ nanosheets (Equation (1)). However, the TiO_2_ peaks could not be clearly identified in the MnO_2_-TiO_2_-carbon-47.28% and 37.81% materials because there was less TiO_2_ content, and they are almost completely covered by intensively immobilized MnO_2_ nanosheets [32].

The carbon and MnO_2_ contents in the resultant MnO_2_-based carbon composites were evaluated using TGA measurements. As exhibited in Figure 5c, below 250 °C, about 8–10 wt % weight loss of the samples occurred due to the evaporation of water and other small molecules. Rapid weight losses during the oxidation of the carbon nanofibers between 300–500 °C in air were calculated as 43.28, 52.72, 51.67 and 59.43 wt % for the MnO_2_-TiO_2_-carbon-47.28%, 37.81% and MnO_2_-carbon-40.15%, 33.30%, respectively. Moreover, the contents of TiO_2_ were too low and could be ignored in the two titania-based composites. Thus, the contents of MnO_2_ contained in the MnO_2_-TiO_2_-carbon-47.28%, 37.81% and MnO_2_-carbon-40.15%, 33.30% are 47.28, 37.81, 40.15 and 33.30 wt %, respectively.

The Mn 2p peaks from the XPS spectrum were tested to study the valence state of Mn in the MnO_2_-TiO_2_-carbon-47.28% composite, and the results are shown in Figure 5d. Two peaks centered at 653.8 and 642.1 eV were indexed to Mn 2p_1/2_ and Mn 2p_3/2_, respectively. The energy separation value of 11.7 eV is well in accordance with the known data for MnO_2_ [37]. Moreover, the oxidation state of Mn in the MnO_2_-TiO_2_-carbon-37.81% composite showed the similar results to those of MnO_2_-TiO_2_-carbon-47.28% composite, which demonstrated the existence of MnO_2_ in the sample (Appendix A).

Figure 6 displays the N_2_ adsorption–desorption isotherms and the pore-size distribution curves of the MnO_2_-TiO_2_-carbon-47.28%, -37.81% composites. Both of the composite materials clearly show type IV isotherms with a type H3 hysteresis loop, indicating the existence of mesopores in the composites [38]. The specific surface areas of the MnO_2_-TiO_2_-carbon-47.28% and 37.81% were 83.2 and 67.1 m^2^ g^−1^ (Figure 6a), and the pore volumes of the two composites were calculated to be 0.299 and 0.238 cm^3^ g^−1^, respectively. A mesoporous structure was observed for the MnO_2_-TiO_2_-carbon-47.28% composite with average sizes of 3.5 and 8.8 nm (Figure 6b). However, the average pore size of the MnO_2_-TiO_2_-carbon-37.81% was measured to be 1.9 nm. Due to the larger specific surface area and the mesoporous structure of the MnO_2_-TiO_2_-carbon-47.28% composite, this material could supply sufficient active sites and accelerate electron transportation during Li^+^ insertion/extraction processes, thus leading to better electrochemical performance of the electrode material. Hence, it is concluded that the composite with a larger pore volume and higher specific surface area is able to deliver a high specific capacity and long cycling life during the charge/discharge processes [31].

### 3.2. Electrochemical Study of the MnO_2_-TiO_2_-Carbon and MnO_2_-Carbon Composites

The electrochemical performances of the nanofibrous MnO_2_-TiO_2_-carbon composites employed as anodic materials for LIBs were tested to evaluate the kinetics of Li^+^ intercalation/deintercalation processes. Figure 7 shows the cyclic voltammetric (CV) curves of MnO_2_-TiO_2_-carbon-47.28%, MnO_2_-carbon-40.15% and MnO_2_-NPs electrodes during charge/discharge processes in the voltage range from 0.01 to 3.0 V with a scan rate of 0.2 mV s^−1^. In the initial cathodic curves of both MnO_2_-TiO_2_-carbon-47.28% and MnO_2_-carbon-40.15% electrodes (Figure 7a,b), a broad peak centered at 0.1 V was attributed to the formation of a solid–electrolyte interface (SEI) layer and a reduction of Mn^2+^ to metallic Mn^0^ [16], and two mild peaks at 1.17 and 2.46 V corresponded to the reduction of MnO_2_ to Mn^2+^ and the decomposition of electrolytes [19,39]. The whole discharge process was calculated as Equation (2):MnO_2_ + 4Li^+^ + 4e^−^ → 2Li_2_O + Mn(2)

Two oxidation peaks at 1.24 and 2.45 V were observed in the first anodic curve, which were attributed to the oxidations of Mn^0^ to Mn^2+^ and Mn^2+^ to Mn^4+^, respectively [39]. The whole charge process was calculated as Equation (3):2Li_2_O + Mn → MnO_2_ + 4Li^+^ + 4e^−^(3)

In the next cycles, the reduction peak at 0.1 V and the oxidation peak at 1.18 V shifted to 0.15 and 1.24 V, respectively. The distinct shifts in the subsequent cycles were attributed to the structural changes and reconstitutions initiated by the formation of amorphous Li_2_O and Mn^0^ [40,41,42,43]. For MnO_2_-NPs, an acuminated cathodic peak centered at 0.08 V and a relatively weaker peak at 1.14 V in the first charge curve were ascribed to the formation of a SEI layer and the reduction of MnO_2_ to Mn (Figure 7c). Two oxidation peaks at 1.24 and 2.47 V were noticed in the charge curves, which meant a two-step electrochemical reaction [19]. The weak redox peaks in the first cycles of the three anodes located at 0.67 and 2.47 V were attribute to the conversion reaction between the Cu metal (copper foam) and CuO, and the Cu_2_O hybrid nanocomposite, and the cathodic peak shift to 0.76 V in subsequent cycles [44]. The peak intensities of the CV curves of MnO_2_-carbon-40.15% and MnO_2_-NPs decreased over the next three charge/discharge cycles, indicating fading of the irreversible capacity of the first cycle. In comparison, the peak intensities of the curves (Figure 7a) overlapped well in the next three cycles, implying excellent electrochemical reversibility and the structural stability of the MnO_2_-TiO_2_-carbon-47.28% anode. The ultra-thin titania film coating layer served as a protective barrier during the reaction of MnO_4_^−^ with carbon nanofiber and led to the uniform immobilization of MnO_2_ nanosheets with higher contents, which was well in accordance with the TGA measurements. The CV curves of the MnO_2_-TiO_2_-carbon-37.81% and MnO_2_-carbon-33.30% anodes showed similar peak shifts as MnO_2_-TiO_2_-carbon-47.28% and MnO_2_-carbon-40.15%, implying analogous Li^+^ insertion/extraction processes (Appendix A). However, the intensities of the charge/discharge curves decreased quickly in subsequent cycles, indicating that the fast irreversible capacity was fading and that there was poor cycling stability.

The charge/discharge curves of the MnO_2_-TiO_2_-carbon-47.28%, MnO_2_-carbon-40.15%, TiO_2_-carbon nanofiber and MnO_2_-NPs electrodes are shown in Figure 8. The first discharge and charge capacities of MnO_2_-TiO_2_-carbon-47.28% composite were 1341 and 547 mAh g^−1^ (Figure 8a), respectively, showing a Coulombic efficiency of 41%. The huge irreversible capacity degradation was due to the formation of a SEI layer that covered the surface of the electrode material [45]. In following cycles, the charge/discharge capacities of the electrode materials were reduced from 556 to 489 mAh g^−1^ as they cycled to the 5th cycle. The Coulombic efficiency increased to 98% at the 10th cycle, the cycling capacities continued to slowly rise and finally reached a constant capacity of 652 mAh g^−1^ by the 100th cycle. A flat voltage plateau at around 0.4–0.5 V during the discharge process indicated the reduction of MnO_2_ to Mn in one step. However, a sharp plateau and a short plateau were observed in following discharge processed, implying two steps of reduction of Li with MnO_2_. Two charge plateaus at around 1.0–1.3 V and 2.2–2.5 V demonstrated the oxidation of Mn^0^ in two steps [33,46]. These results are well in accordance with the CV measurements. As a comparison, the MnO_2_-TiO_2_-carbon-37.81% electrode exhibited discharge and charge capacities of 1259 and 477 mAh g^−1^, respectively, resulting in a Coulombic efficiency of 37.9%. Moreover, the discharge capacity fell to 407 mAh g^−1^ after 10 cycles and slowly increased to a stable capacity of 494 mAh g^−1^ by the 100th cycle (Appendix A). The MnO_2_-TiO_2_-carbon-47.28% composite showed better electrochemical performance than MnO_2_-TiO_2_-carbon-37.81%, and its charge/discharge plateaus were longer, which implied well-organized MnO_2_ nanosheets with higher contents in the composite-47.28% electrode that furnished adequate active sites for the Li^+^ and electron transport. The discharge capacities of MnO_2_-carbon-40.15% electrodes dropped rapidly from 1251 to 367 mAh g^−1^ when cycled for 20 cycles (Figure 8b). The large capacity loss of the MnO_2_-carbon-40.15% electrode was related to the weak adhesion between the MnO_2_ nanosheets and the carbon nanofiber, thus leading to the exfoliation and structure destruction of MnO_2_ during cycling. As for the MnO_2_-carbon-33.30% electrodes, the primary discharge and charge capacities of the composite were 1053 and 386 mAh g^−1^, respectively, presenting a Coulombic efficiency of 36.7% (Appendix A). The severe capacity degradation of the MnO_2_-carbon-33.30% was attributed to the weak adhesion between the MnO_2_ nanosheets and carbon nanofibers that led to the debonding of MnO_2_ nanosheets from the carbon nanofibers. These exfoliated MnO_2_ nanosheets may have suffered from huge volume changes and lost their effective contact with the electrolyte, resulting in the poor electrochemical performance of the electrodes [34]. The discharge capacity the TiO_2_-carbon nanofibers decreased from 985 to 249 mAh g^−1^ after 100 cycles (Figure 8c). This further demonstrated that the immobilized MnO_2_ nanosheets significantly boosted the capacity of MnO_2_-TiO_2_-carbon composites and improved the structure stability of the electrodes during the Li^+^ intercalation/deintercalation processes. However, the initial discharge and charge capacities of MnO_2_-NPs were 1276 and 464 mAh g^−1^, respectively, giving a Coulombic efficiency of 36.4% (Figure 8d), and the capacity steadily decreased to 146 mAh g^−1^ after 100 cycles. The great irreversible capacity loss was ascribed to the aggregation and pulverization of MnO_2_-NPs during the repeated charge/discharge processes, thus leading to poor cycling stability as well as a low rate capability.

The cycling performance of the MnO_2_-TiO_2_-carbon-47.28%, 37.81%, MnO_2_-carbon-40.15%, 33.30%, TiO_2_-carbon nanofibers and MnO_2_-NPs are shown in Figure 9a. For the MnO_2_-TiO_2_-carbon-47.28% composite, the capacity dropped rapidly in the first three cycles and thereafter increased gradually after 10 cycles, reaching a capacity of 677 mAh g^−1^ by the 130th cycle, delivering a Coulombic efficiency of over 99%. The reason for the improvement in capacity has been elucidated for many transition metal oxides, which might be attributed to the reversible growth of the polymeric gel-like film and the activation of electrode materials during Li^+^ insertion/extraction processes [46,47,48]. A variety of reported MnO_2_ nanostructures and MnO_2_-based composites have turned out to be good anode materials for LIBs [18,33,34,49,50,51]. The present fabricated bio-inspired nanofibrous MnO_2_-TiO_2_-carbon-47.28% composite showed comparable or better capacity capability when tested at identical current densities (Table 1). These enhancements were ascribed to the three-dimensional reticular structure and the high surface-to-volume ratio of the composite derived from the natural cellulosic substances, which effectively alleviated the large volume expansion and exfoliation of the MnO_2_ nanosheets. Therefore, the MnO_2_-TiO_2_-carbon-47.28% electrode material exhibited superior electrochemical performance during cycling. However, the MnO_2_-TiO_2_-carbon-37.81%, with a relative lower content of MnO_2_, showed a stable capacity of 491 mAh g^−1^ after 130 cycles, implying that the higher MnO_2_ loading is capable of contributing a larger capacity to the composites. As a comparison, the capacity of MnO_2_-carbon-40.15% faded sharply in the initial 20 cycles and a steady capacity of 425 mAh g^−1^ was obtained after 130 charge/discharge cycles. The capacity difference among the MnO_2_-TiO_2_-carbon and MnO_2_-carbon-40.15% composites suggested that the ultrathin titania film pre-coated on the surface of the carbon nanofibers plays an important role in improving the stability of the cycling performance of the electrodes. It was found that the TiO_2_ coating layer facilitated the growth of well-organized MnO_2_ nanosheets with a higher content in the case of the KMnO_4_ precursor with a high concentration is used. While the discharge capacity of MnO_2_-carbon-33.30% dropped from 1053 to 335 mAh g^−1^ after 130 cycles. This further illustrated that MnO_2_ nanosheets were likely to be peeled off from the composite and tended to aggregate during Li/Li^+^ processes, thus leading to severe capacity fading. The TiO_2_-carbon nanofibers delivered a fast capacity loss in the first 20 cycles and slowly decreased to 235 mAh g^−1^ by the 130th cycle, corresponding to a 58.5% retention of the first charge capacity. The large irreversible capacity degradation of the electrode was ascribed to the formation of a SEI layer. For the MnO_2_-NPs, a sudden capacity loss was observed in the initial 30 cycles and then slowly reduced to 145 mAh g^−1^ at the 130th cycle, resulting in a capacity retention of 29.8% for the second discharge capacity. This phenomenon can be explained by the huge volume expansion and aggregation of MnO_2_-NPs during the electrochemical processes.

To evaluate the capacity contribution of the MnO_2_ component to MnO_2_-TiO_2_-carbon-47.28%, 37.81% and MnO_2_-carbon-40.15%, 33.30% composites, the capacities of MnO_2_ were calculated using the following Equation (4):$$C\;(\mathrm{total})=C_{\left({\mathrm{TiO}}_{2}/\mathrm{carbon}\right)}\;\times \;W_{\left(\mathrm{carbon}\right)}\;\mathrm{wt}\%+C_{\left({\mathrm{MnO}}_{2}\right)}\times \;W_{\left({\mathrm{MnO}}_{2}\right)}\;\mathrm{wt}\%$$(4)

Since the TiO_2_ in the MnO_2_-TiO_2_-carbon composite was too low and could be ignored, the separated capacity contribution of titania and carbon constituents in the electrodes were calculated as the whole capacity of the TiO_2_-carbon. The results are listed in Table 2, where the contributed specific capacities of MnO_2_ in the MnO_2_-TiO_2_-carbon-47.28%, 37.81%, MnO_2_-carbon-40.15%, 33.30% electrodes were 1169.9, 912.5, 708.2, 535.3 mAh g^−1^, respectively. The contribution percentages of MnO_2_ were equivalent to 95.1%, 74.2%, 57.6%, and 43.5% of the theoretical capacity (1230 mAh g^−1^), respectively. Significant improvements to the contributing capacity and percentage of MnO_2_ in the MnO_2_-TiO_2_-carbon-47.28% electrode were ascribed to its three-dimensionally interwoven structures with its large specific area, which enabled faster Li^+^ and electron transport during the electrochemical processes. Moreover, the mesoporous structure with a higher content of MnO_2_ nanosheets in the composite made a large contribution to the enhancement of the capacity. Therefore, the MnO_2_-TiO_2_-carbon-47.28% composite showed superior cycling stability and a high specific capacity when compared with the other three MnO_2_-based counterparts.

To further investigate the electrochemical superiority of the MnO_2_-TiO_2_-carbon-47.28% composite, the rate performances of the composite electrodes at varied current rates are shown in Figure 9b. The MnO_2_-TiO_2_-carbon-47.28% electrode delivered discharge-specific capacities of 491, 406, 312, 232, 158, and 94 mAh g^−1^ with current densities increasing from 100, 200, 500, 1000, 2000, and 4000 mA g^−1^, respectively. When the current rate was set back to 100 mA g^−1^, a reversible capacity of 493 mAh g^−1^ was obtained and successively increased to 560 mAh g^−1^, implying the favorable rate capabilities of the electrode. In the meantime, the rate capability of MnO_2_-TiO_2_-carbon-37.81% was slightly higher than that of the MnO_2_-carbon-40.15%, and the discharge capacities of composite-37.81% were 392, 319, 212, 132, 73, and 44 mAh g^−1^ at current rates of 100, 200, 500, 1000, 2000, and 4000 mA g^−1^, respectively. However, the specific capacities of the MnO_2_-carbon-40.15% electrode were 376, 313, 194, 120, 61, and 41 mAh g^−1^ at comparable current densities for composite-37.81%. Though the content of MnO_2_ in the MnO_2_-TiO_2_-carbon-37.81% was lower than that of MnO_2_-carbon-40.15%, the composite-37.81% still displayed a better rate capability than composite-40.15%. It was concluded that the structural stability of the composite and the strong adhesion of MnO_2_ nanosheets to the titania film pre-coated carbon nanofibers greatly prevented the exfoliation and pulverization of MnO_2_ in MnO_2_-TiO_2_-carbon-37.81%, and led to a superior rate performance. The MnO_2_-carbon-33.30% anode showed lower specific capacities of 368, 281, 184, 115, 56, 37, and 330 mAh g^−1^ at the current rates of 100, 200, 500, 1000, 2000, and 4000 mA g^−1^, respectively. A reversible capacity of 330 mAh g^−1^ was retained when the current rate returned to 100 mA g^−1^. The MnO_2_-carbon-33.30% electrode exhibited a poor rate performance because of less loading content of MnO_2_ nanosheets, as well as the weak force between the carbon nanofibers and the MnO_2_, resulting in the structural destruction of active materials. As for the TiO_2_-carbon nanofibers, the specific capacities were obviously lower than those of the four MnO_2_-based samples mentioned above. In comparison, the specific capacity of MnO_2_-NPs decreased sharply in the initial 10 cycles. When the current rate was set in the range of 500–4000 mA g^−1^, the capacity reduced to as low as 10 mAh g^−1^. An irreversible capacity of 167 mAh g^−1^ was acquired when the cycling density returned to 100 mA g^−1^. This further demonstrated that the severe volume expansion and aggregation of MnO_2_-NPs during the Li^+^ intercalation/deintercalation processes are the main cause of the poor rate capability of the electrode.

To better understand the greatly enhanced electrochemical performances of the MnO_2_-TiO_2_-carbon-47.28% electrode, electrochemical impedance spectroscope (EIS) measurements were tested to compare the conductivity of the prepared MnO_2_-TiO_2_-carbon-47.28%, MnO_2_-carbon-40.15% and TiO_2_-carbon electrodes after 200 repeated cycles. Nyquist plots of the three electrodes as shown in Figure 10 and depict a semicircle in the high frequency zone and an inclined line in the low-frequency region, corresponding to the charge transfer resistance and the Li ion diffusion activity, respectively. The kinetic parameters were evaluated using the equivalent circuit (inset in Figure 10), and *R_s_* represents the internal resistance of the battery, *R_ct_* is the charge transfer impedance on the electrode-electrolyte interface, *W* is the Warburg impedance of Li^+^ diffusion throughout the active materials, and CPE is the constant phase-angle element [35,52]. The fitting results are listed in Appendix A, although the *R_s_* values of the MnO_2_-TiO_2_-carbon-47.28%, MnO_2_-carbon-40.15% and TiO_2_-carbon electrodes showed little difference, the values of *R_ct_* of the three composite electrodes were measured to be 46.32, 155, and 231.7 Ω, respectively. The MnO_2_-TiO_2_-carbon-47.28% had the lowest *R_s_* and *R_ct_* values; this strongly confirmed that the composite-47.28% had the lowest activation energy and underwent a fast charge transfer reaction, leading to a decrease om the internal resistance of the entire battery. In addition, the structural morphology property of the MnO_2_-TiO_2_-carbon-47.28% electrode after 200 charge/discharge cycles was partially retained (Appendix A), demonstrating that structural stabilization was achieved and exhibited excellent cycling stability and reversible rate capability.

## 4. Conclusions

A cellulose substance-derived nanofibrous MnO_2_-TiO_2_-carbon composite was fabricated by immobilizing porous MnO_2_ nanosheets on the surface of titania-coated carbon nanofibers employing a natural cellulose substance as both the carbon source and the structural matrix. The material, with a large surface area, faithfully inherited the three-dimensional network structures of the original cellulose substances. The MnO_2_-TiO_2_-carbon-47.28% composite showed durable cycling performance and stable rate capabilities when employed as anodic materials for LIBs. The improved electrochemical performances of the MnO_2_-TiO_2_-carbon-47.28% are attributed to the synergistic effect of the carbon nanofibers, the ultrathin titania coating layer, as well as the high content of MnO_2_ nanosheets. The internal carbon nanofibers act as a conductive matrix and facilitate Li ion and electron transfer during Li^+^ insertion/extraction processes. Moreover, the titania layer plays a key role in the higher loading of MnO_2_ nanosheets, and provides an intermediate interface for the strong adhesion between the carbon nanofiber and MnO_2_. The mesoporous MnO_2_ nanosheets with a large theoretical specific capacity contribute to the higher capacity retention. The current method developed for the fabrication of bio-inspired electrode material provides a novel path for the introduction of sophisticated nanoarchitectures into energy-related fields.

## Data Availability

The data presented in this study are available on request from the corresponding author.

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
