# Peer review of "A Cellulose-Derived Nanofibrous MnO2-TiO2-Carbon Composite as Anodic Material for Lithium-Ion Batteries"

_materials, 2021, doi:10.3390/ma14123411_

Round 1
Reviewer 1 Report
The manuscript titled “A Cellulose-Derived Nanofibrous MnO2-TiO2-Carbon Compo-site as Anodic Material for Lithium-Ion Batteries” has been well written along with enough characterization results and discussion. Therefore, I can accept the present manuscript to publish in Materials journal. However, prior to the acceptance, the authors should carry out the following corrections in the revised manuscript.
In BET studies, authors mention the specific surface area value. How much the pore volume and surface area influence the specific capacity and capacity retention?
How authors optimize the specific surface area and porosity?
Give any comparison studies or table of previous literature concern about MnO2-TiO2-Carbon synthesis and LIB studies (specific capacity; electrolyte and stability for number of cycles).
Give crystallographic or microscopic evidence of MnO2-TiO2-Carbon after 200 cycles. Check the crystal structure after stability studies. Justify?
English and grammar should be check with native English speaker.
In introduction section the sentence starts with, compared with conventional commercialized graphite anodes, transition metal oxides……………...durable discharging rates. Authors should add the following recent references.
Energy Fuels 34(11) (2020) 14958–14967.
Ionics 26 (2020) 345–354.
Journal of Energy Storage 26 (2019) 100914.
Energy Storage 30 (2020) 101483.
Apart from these, I have no major concerns about this interesting paper, and I recommend its publication with minor revision.
Author Response
Manuscript ID: materials-1261241 for Materials
Response to the comments of Reviewer #1
Overall comments: The manuscript titled “A Cellulose-Derived Nanofibrous MnO2-TiO2-Carbon Composite as Anodic Material for Lithium-Ion Batteries” has been well written along with enough characterization results and discussion. Therefore, I can accept the present manuscript to publish in Materials journal. However, prior to the acceptance, the authors should carry out the following corrections in the revised manuscript.
Answer: The reviewer thinks that the manuscript is a well-written article along with enough characterization results and discussion, and he/she can accept the present manuscript to publish in Materials Journal after some minor corrections that are considered and revised. We sincerely thank the reviewer for his/her valuable comments on our work, which are helpful for us to improve the work. And we have thoroughly revised the manuscript according to his/her comments.
Comment #1. In BET studies, authors mention the specific surface area value. How much the pore volume and surface area influence the specific capacity and capacity retention?
Answer: We thank the reviewer for this valuable comment. The pore volumes of the MnO2-TiO2-carbon-47.28% and 37.81% are measured to be 0.299 and 0.238 cm3 g−1, respectively. Due to the higher specific surface area and the mesoporous structure of the MnO2-TiO2-carbon-47.28% composite, this material can provide more active sites and facilitate the electron transfer during the Li+ insertion/extraction processes, thus leading to the better electrochemical performance of the electrode material. Hence, It is concluded that the composite with larger pore volume and higher specific surface area is able to deliver high specific capacity and long cycling life during the charge/discharge processes (Chem Asian J. 2021, 16, 378). Corresponding discussions and references have been newly added in the revised manuscript (Page 10, Lines 20 to 21; Page 11, Lines 6 to 8).
Comment #2. How authors optimize the specific surface area and porosity?
Answer: We appreciate the reviewer for this valuable suggestion. Actually, the specific surface area and porosity of the MnO2-based composites are not optimized. In our previous work, the specific surface area and the pore volume of the titania-coated carbon, a nanofibrous material derived from natural cellulose substance (e.g., ordinary laboratory filter paper), were calculated to be 404 m2 g−1 and 0.23 cm3 g−1, respectively. The porous titania-carbon hybrid material possesses a rather narrow pore size distribution centered at 3.9 nm (Chem. Eur. J. 2010, 16, 7730). In spite of the densely immobilization of the porous MnO2 nanosheeets, the specific surface area of the current MnO2-TiO2-carbon-47.28% composite is still as large as 83.2 m2 g−1. Moreover, the average sizes of the composite are 3.5 and 8.8 nm. This further demonstrates the MnO2-TiO2-carbon-47.28% composite with large specific surface area and high porosity facilitates the Li+ and e- transfer as well as buffers the huge volumetric expansion of MnO2 during the charge/discharge processes, thus leading to the excellent electrochemical performances of the electrode materials.
Comment #3. Give any comparison studies or table of previous literature concern about MnO2-TiO2-Carbon synthesis and LIB studies (specific capacity; electrolyte and stability for number of cycles).
Answer: We thank the reviewer for the professional suggestion. The comparisons of the electrochemical properties (specific capacity; electrolyte and stability for number of cycles) of the MnO2-TiO2-Carbon and other MnO2-based nanocomposites are summarized in Table 1, and corresponding discussion has been newly added in the revised manuscript (Page 14, Table 1; Page 14, Lines 13 to 19; Page 15, Lines 1 to 3).
Comment #4. Give crystallographic or microscopic evidence of MnO2-TiO2-Carbon after 200 cycles. Check the crystal structure after stability studies. Justify?
Answer: We thank the reviewer for this comments. We feel so regret that we have not test the crystal structure of the MnO2-TiO2-carbon composite after stability studies due to the difficulty in obtaining sufficient target anode materials in the type coin cells. However, the TEM image (Figure S7, see the Supplementary Materials) of the MnO2-TiO2-carbon-47.28% anode material after 200 charge/discharge cycles shows that the structural morphology property of composite electrode after 200 charge/discharge cycles is partially retained, demonstrating the structural stabilization is achieved and exhibits excellent cycling stability and reversible rate capability. (Page 17, Lines 18 to 21).
Comment #5. English and grammar should be check with native English speaker.
Answer: We thank the reviewer for this suggestion. The English and grammar of this manuscript has been checked and revised carefully.
Comment #6. In introduction section the sentence starts with, compared with conventional commercialized graphite anodes, transition metal oxides……………...durable discharging rates. Authors should add the following recent references. Energy Fuels 34(11) (2020) 14958–14967. Ionics 26 (2020) 345–354. Journal of Energy Storage 26 (2019) 100914 Energy Storage 30 (2020) 101483.
Answer: We thank the reviewer for this good suggestion and bringing our attention to these literatures. The references have been newly cited in the specific place to support the statement in the Introduction section as suggested (Page 2, Line 14, Refs. 8-11).
------------------------------------------------------------------------------------
Manuscript ID: materials-1261241 for the Materials
List of the changes made in the manuscript according to the comments of Reviewer #1
- Page 2, Line 14: 8, 9, 10, 11 are newly cited in the specific place to support the statement in the Introduction section as suggested (according to comment 6)
- Page 10, Lines 20 to 21; Page 11, Lines 6 to 8: The pore volumes of the MnO2-TiO2-Carbon composite are added, and the discussions about the influence of the pore volume and surface area to the specific capacity and capacity retention with newly cited 31 are also added. (according to comment 1)
- Pages 14, Lines 13 to 19, Table 1; Page 15, Lines 1 to 3: The Table 1 is added to compare the electrochemical properties of the MnO2-TiO2-Carbon and other MnO2-based nanocomposites, and corresponding discussion is also added. (according to comment 3)

Reviewer 2 Report
The authors investigate a bio-inspired composite as anodic material for Li-ion batteries. Cycling stability is shown to be improved. The work is generally interesting. There are several English mistakes and confusing sentences that need addressing. (for instance, in the third row of the conclusions celllulose). Aside from this, I think the paper can be accepted once the language is sharpened.
Author Response
Manuscript ID: materials-1261241 for Materials
Response to the comments of Reviewer #2
Overall comments: The manuscript has a topical subject for a more green energy and the paper deserves publication in journal Materials being well written with properly selected new references in both field of materials and energy. The methodology is rich and suitable for the topics and generally the manuscript is interesting for many groups of readings. Before publication in my opinion is a need for revision taking into account the following modifications.
Answer: We highly appreciate the valuable suggestions made by the reviewer. He/She thinks that the paper has a topical subject for a more green energy and the paper deserves publication in journal Materials. The manuscript is being well written with properly selected new references in both field of materials and energy. Moreover, the methodology is rich and suitable for the topics and generally the manuscript is interesting for many groups of readings. We sincerely highly appreciate the approval to our work and the valuable suggestions made by the reviewer, which are beneficial for us to improve the work. And we have thoroughly revised the manuscript according to his/her comments.
Comment #1. To present more clearly in the introduction the novelty of the manuscript.
Answer: We thank the reviewer for this valuable comment. The introduction of the novelty of the current work has been newly added in the revised manuscript (Page 3, Lines 8 to 14).
Comment #2. To improved the results organization which now has 10 figures and only one table.
Answer: We thank the reviewer for this comment. The manuscript is submitted to journal Materials as a full paper. According to Reviewer 1, one more Table concerning the comparisons of the electrochemical properties (specific capacity; electrolyte and stability for number of cycles) of the MnO2-TiO2-Carbon and other MnO2-based nanocomposites has been newly added as new Table 1, and corresponding discussion has been added in the revised manuscript (Page 14, Table1; Page 14, Lines 13 to 19; Page 15, Lines 1 to 3).
Comment #3. To look more carefully to the references subchapter which references generally properly chosen, but a part of them have a better place in introduction (see 47-51). Also the number of self citations seems to me that are more than 20%.
Answer: We greatly thank the reviewer for his/her careful review on our paper. The references 47-51 have been placed in the Introduction part (Page 3, Line 8). Moreover, in order to avoid too many self citations, the Refs. 20, 26, 27, 47, 49, 51 have been deleted.
---------------------------------------------------------------
Manuscript ID: materials-1261241 for the Materials
List of the changes made in the manuscript according to the comments of Reviewer #2
- Page 3, Lines 8 to 14: The introduction of the novelty of the current work with newly cited 31 is added. (according to comment 1)
- Page 2, Line 49; Page 3, Line 8, Line 49; Page 18, Line 4: The 47-51 have been placed in the Introduction part (Page 3, Line 8); In order to avoid too many self citations, the initial Refs. 20, 26, 27,47, 49, 51 have been deleted, the refrences are re-numbered accordingly. (according to comment 3)
- Pages 14, Lines 13 to 19, Table 1; Page 15, Lines 1 to 3: The Table 1 is added to compare the electrochemical properties of the MnO2-TiO2-Carbon and other MnO2-based nanocomposites, and corresponding discussion with newly cited 50, 51, 52 is also added. (according to comment 2)

Reviewer 3 Report
The manuscript has a topical subject for a more green energy and the paper deserves publication in journal Materials being well written with properly selected new references in both field of materials and energy.
The methodology is rich and suitable for the topics and generally the manuscript is interesting for many groups of readings.
Before publication in my opinion is a need for revision taking into account the following modifications:
a)Fistly I do strongly believe that in our time when the number of papers having the same approach for a specific subject is very large it is very important to explained with details what is really new in a manuscript comparing with existing literature (as example to compare with reference 12)
b) it is mandatory to better organized material with more tables and I do consider that this fact is in the benefit of paper scientific quality introducing more quantified data.
c) the number of self citations is important in a manuscript and of course is a way to present the authors expertise in the field , but much more that 20%, such as in the present paper is not in the paper benefit.
Author Response
Manuscript ID: materials-1261241 for Materials
Response to the comments of Reviewer #3
Overall comments: The manuscript has a topical subject for a more green energy and the paper deserves publication in journal Materials being well written with properly selected new references in both field of materials and energy. The methodology is rich and suitable for the topics and generally the manuscript is interesting for many groups of readings. Before publication in my opinion is a need for revision taking into account the following modifications.
Answer: The reviewer thinks that our paper is interesting for many groups of readings and recommends its publication in the Materials with some modifications. We sincerely thank the reviewer for his/her approval of our work.
Comment #1. Fistly I do strongly believe that in our time when the number of papers having the same approach for a specific subject is very large it is very important to explained with details what is really new in a manuscript comparing with existing literature (as example to compare with reference 12).
Answer: We thank the reviewer for this comment. The description of the novelty of the current work has been newly added in the revised manuscript (Page 3, Lines 8 to 14) with comparasion with Ref. 16 ( which is Ref. 12 in the original version).
Comment #2. It is mandatory to better organized material with more tables and I do consider that this fact is in the benefit of paper scientific quality introducing more quantified data.
Answer: We thank the reviewer for this comment. A new table (Page 14, Table 1) has been added to give a comparasion of our results with those in the literatures, and the corresponding discussion has also been added (Page 14, Table1; Page 14, Lines 13 to 19; Page 15, Lines 1 to 3).
Comment #3. The number of self citations is important in a manuscript and of course is a way to present the authors expertise in the field, but much more that 20%, such as in the present paper is not in the paper benefit.
Answer: We greatly thank the reviewer for the careful screening of our paper. In order to avoid too many self citations, the original Refs. 20, 26, 27, 47, 49, 51 have been deleted.
---------------------------------------------------------------------
Manuscript ID: materials-1261241 for the Materials
List of the changes made in the manuscript according to the comments of Reviewer #3
- Page 3, Lines 8 to 14: The description of the novelty of the current work with a comparasion with the literature is added. (according to comment 1)
- Pages 14, Lines 13 to 19, Table 1; Page 15, Lines 1 to 3: A new table (Table 1) has been added to give a comparasion of our results with those in the literatures, and the corresponding discussion has also been added with newly cited 50, 51, 52. (according to comment 2)
- In order to avoid too many self citations, the original Refs. 20, 26, 27, 47, 49, 51 have been deleted. (according to comment 3)
